# Sex-Biased Expression and Response of microRNAs in Neurological Diseases and Neurotrauma

**DOI:** 10.3390/ijms25052648

**Published:** 2024-02-24

**Authors:** Urim Geleta, Paresh Prajapati, Adam Bachstetter, Peter T. Nelson, Wang-Xia Wang

**Affiliations:** 1Sanders-Brown Center on Aging, College of Medicine, University of Kentucky, Lexington, KY 40536, USA; ugge222@uky.edu (U.G.); paresh.prajapati@uky.edu (P.P.); adam.bachstetter@uky.edu (A.B.); pnels2@uky.edu (P.T.N.); 2Spinal Cord and Brain Injury Research Center, College of Medicine, University of Kentucky, Lexington, KY 40536, USA; 3Neuroscience, College of Medicine, University of Kentucky, Lexington, KY 40536, USA; 4Pathology and Laboratory Medicine, College of Medicine, University of Kentucky, Lexington, KY 40536, USA

**Keywords:** sex biased, microRNA, neurodegenerative diseases, traumatic brain injury, X chromosome, miR-223-3p

## Abstract

Neurological diseases and neurotrauma manifest significant sex differences in prevalence, progression, outcome, and therapeutic responses. Genetic predisposition, sex hormones, inflammation, and environmental exposures are among many physiological and pathological factors that impact the sex disparity in neurological diseases. MicroRNAs (miRNAs) are a powerful class of gene expression regulator that are extensively involved in mediating biological pathways. Emerging evidence demonstrates that miRNAs play a crucial role in the sex dimorphism observed in various human diseases, including neurological diseases. Understanding the sex differences in miRNA expression and response is believed to have important implications for assessing the risk of neurological disease, defining therapeutic intervention strategies, and advancing both basic research and clinical investigations. However, there is limited research exploring the extent to which miRNAs contribute to the sex disparities observed in various neurological diseases. Here, we review the current state of knowledge related to the sexual dimorphism in miRNAs in neurological diseases and neurotrauma research. We also discuss how sex chromosomes may contribute to the miRNA sexual dimorphism phenomenon. We attempt to emphasize the significance of sexual dimorphism in miRNA biology in human diseases and to advocate a gender/sex-balanced science.

## 1. Introduction

There are substantial neurobiological differences between males and females during development, adulthood, and senescence [1,2,3,4,5,6,7,8,9,10]. As such, it is predictable that numerous neurological and age-related diseases have distinct prevalence, progression, and therapeutic responses between genders. For instance, women have a significantly increased risk of developing Alzheimer’s disease (AD) relative to men [11,12,13,14]. According to Alzheimer’s Disease Facts and Figures from the Alzheimer’s Association (https://www.alz.org/media/Documents/alzheimers-facts-and-figures.pdf (accessed on 20 February 2024)), almost two-thirds of Americans living with AD are women. Additionally, the prevalence of multiple sclerosis (MS) is four times higher in women compared to men [15,16]. By contrast, there is a significantly (~1.5×) higher incidence among men for Parkinson’s disease (PD) [17]. Amyotrophic lateral sclerosis (ALS) represents another sexual dimorphic neurodegenerative disease, with the incidence among men being three times higher than among women, and men typically experience an earlier onset of the disease [18,19]. Neuropsychiatric disorders also exhibit substantial sex bias in prevalence, symptoms, and age of onset [20]. For example, men have a higher prevalence of schizophrenia, hyperactivity disorder, and autistic spectrum, and women have a higher prevalence of anxiety disorders and depression [21].

The pathogenesis of neurological diseases is associated with multifactorial contributing factors, including genetic predisposition, inflammation, and environmental exposures. Sex differences in neurological diseases may be associated with any or all of these influences. Research indicates that while the increased risk of developing AD associated with the *APOE E4* allele is generally assumed to be equal in men and women, this risk is actually stronger in women [22,23]. In a Korean genome-wide study on PD, the risk variants *SNCA* and *PARK16* exhibited sex-specific patterns: *SNCA* locus single nucleotide polymorphisms (SNPs) predominated in females, whereas *PARK16* locus SNPs were greater in males [24].

Sex is not only associated with different disease epidemiology but also impacts outcomes and responses to treatment. For example, following a stroke, women experience worse outcomes and undergo more functional decline than men [25,26]. Huntington’s disease (HD), which is highly genetically determined with limited therapeutic options [27], has a higher incidence and faster progression in women compared to men [28,29,30]. Moreover, sexually dimorphic responses to environmental modulators are also noted in HD [27,29]. Traumatic brain injury (TBI) represents another neurological condition where sex differences impact the outcomes, responses to treatment, and recovery. According to the Centers for Disease Control and Prevention (CDC), men are two-times more likely to be hospitalized and three-times more to die due to TBI, yet women were more vulnerable than men to show persisting TBI-related cognitive and somatic symptoms [31,32,33,34]. Moreover, a longer time to symptom resolution was reported in high school and collegiate female athletes [35,36,37,38].

Although a myriad of sex-specific genes and pathways have been identified, the mechanisms underlying the sexual dimorphism in diseases and pathological conditions are quite incompletely understood. As a class of powerful biological regulators in all known metazoan and plant species, microRNAs (miRNAs) have emerged as an important regulator of sexually dimorphic responses. However, few studies have examined the extent to which miRNAs contribute to the sex-biased differences observed in various neurological diseases. Here, we review the current state of knowledge related to miRNA sexual dimorphism in neurological diseases and neurotrauma research. We also discuss how sex chromosomes may contribute to the miRNA sexual dimorphism phenomenon. With this review, our goal is to emphasize the significance of sexual dimorphism in miRNA biology in human diseases and to enrich the framework for upcoming studies.

## 2. Overview of microRNAs

MiRNAs are short non-coding RNAs (19–25 nt) that regulate protein translation, usually by inhibiting translation or destabilizing the messenger RNA (mRNA) of a targeted gene. Since the discovery of first miRNA in *Caenorhabditis elegans* [39,40], phylogenetically conserved miRNAs have been characterized across many animals and plants, and the key roles of miRNAs in gene regulation are well recognized. It is estimated that at least 60% of human coding genes are regulated by miRNAs [41]. MiRNAs regulate diverse biological processes such as development, cell differentiation, proliferation, apoptosis, neurodevelopment, senescence, and immunity, and are an integral regulatory component of biological and pathological systems [42,43,44,45,46,47,48,49,50,51].

The biogenesis of miRNA involves complex processes with a stereotypical pathway that flows through multiple protein complexes [44,52]. MiRNAs are typically transcribed as primary miRNA transcripts (Pri-miRNAs) from either the introns of their host genes or standalone miRNA genes by RNA polymerase II. Pri-miRNA is further processed in the nucleus by the Drosha/DGCR8 protein complex to a 70 bp stem-loop precursor-miRNA (pre-miRNA) [44,53,54]. Pre-miRNAs are then exported to the cytoplasm by the cotransporter Exportin 5 and then further cleaved by Dicer into small double-stranded RNA in the cytoplasm [44,55,56,57]. One of the two double strands determined to be the “guide strand” is then loaded into the Argonaute (AGO)-containing RNA-induced silencing complex (RISC) [58,59], while the other strand (passenger strand) is degraded. The AGO-containing miRNA RISC complex is the effector unit of miRNA regulation, in which different members of the AGO paralogs execute overlapping and distinct actions during gene suppression [60]. For example, AGO1 mediates alternative splicing via interaction with cell-type-specific transcriptional enhancers [61]. AGO2- and AGO3-RISC complexes possess a ‘slicer’ function that may directly lead to mRNA degradation [60,62,63].

The complex processes of miRNA biogenesis and function, and the genes/proteins involved are also subject to a spectrum of regulatory mechanisms, which include transcription regulation, developmental staging, pathological stimuli, and sex hormones [64,65,66]. In terms of the widespread regulation of miRNAs by sex hormones, which is not the main focus of the current review, we recommend several excellent early review articles [67,68,69,70].

Some miRNA-related processing mechanisms are sexually dimorphic. For example, AGO4 localized in mammalian germ cells plays a key role in regulating meiotic sex chromosome inactivation [71]. Deficiency of AGO4 in germ cells caused a significant reduction in miRNAs, in which over a 20% loss of miRNAs were X chromosome-linked miRNAs (X-linked miRNAs) [71]. Additionally, miRNAs are differentially expressed in males and females across cells, tissues, organs, and extracellular circulation, and are associated with sex-biased biological and pathological processes [72,73,74,75,76,77,78,79,80,81,82,83,84,85,86,87,88,89,90,91,92,93,94,95,96,97,98]. However, the mechanisms for the sexually dimorphic expression of miRNAs and their biological impact in both normal and pathological events are largely unexplored [67].

## 3. Sex-Biased miRNAs in Neurodegenerative Diseases and Neuropsychiatric Disorders

Differential expressions and regulation of protein-coding genes by miRNAs between males and females may represent the key mechanisms underlying the sex-biased prevalence, pathogenesis, response to therapy, and outcomes of human diseases. As an important class of gene expression regulators, miRNAs play critical roles in neuronal development, differentiation, and brain morphogenesis [49,99,100]. This implies that their sexually dimorphic expression might influence the risk of neurological diseases from early development stages. A study revealed that the expression of the miR-200 family of miRNAs (miR-141-3p, miR200a-3P, miR200b-3P, and miR-429) as well as miR-875 showed a sex-specific switch in the developing rat brain [85]. At postnatal day 0 (P0), these miRNAs were elevated in females; at P7 and the adult stage, all these miRNAs were elevated in male brains (Table 1). Additionally, miRNA may influence response to early life events, such as environmental stresses, which may impact the development of neurological disease later in life.

Sex-biased miRNA expression at different stages of development may lead to gender-specific changes in the onset, progression, and outcome of a neurological disease, especially neuropsychiatric disorders [101]. Prenatal environmental stresses (e.g., maternal stress, diet, and endocrine disruption) can have a major impact on brain development and contribute to the development of neuropsychiatric disorders, such as depression, schizophrenia, and autistic spectrum disorders [102,103,104,105]. These adverse effects are often sex-biased and can persist into subsequent generations [106,107,108,109]. Epigenetic mechanisms [110,111,112,113] and miRNA dysregulation [79,80], among others, were suggested to be associated with these phenomena. It was shown that early gestation is a perinatal period when organizational gonadal hormones establish the sexually dimorphic brain and are most susceptible to maternal stress [80,114]. In a mouse model of demasculinization, a study showed the paternal transmission and programming of the prenatal stress-induced demasculinized phenotype in second-generation (F2) male offspring [80]. The miRNA analysis using the brains of these male offspring 1 day after birth revealed a significant reduction in miR-322, miR-574, and miR-873, with levels similar to those of control female brains. The reduction in these miRNAs was correlated with the significant increase in β-glycan, a common target of all three miRNAs. β-glycan is a member of the TGFβ superfamily and it may be involved in regulating the release of gonadal hormones [80]. Another study showed that estrogen-sensitive miR-30b was found to be significantly reduced in the cerebral cortex of female but not male subjects with schizophrenia [96]. This sex-biased miRNA expression appeared to be regulated by estrogen and estrogen receptor signaling.

MiRNAs directly regulate key molecules in the pathogenesis of various neurological diseases, and numerous dysregulated miRNAs have been characterized [115,116,117,118,119,120]. Unfortunately, few studies have considered sex as a biological factor, leaving the role of sex-biased miRNAs in neurological diseases and neurotrauma largely undefined.

Certain trends have been identified among the limited studies report sex-specific miRNA expression in injured and diseased brains (Table 1). For instance, in a tau pathology mouse model (PS19), miRNAs were considerably more altered in males compared to females [91]. Another study completed by Chum et al. investigated 599 miRNAs in brain vessels extracted from male and female 3xTg-AD mice [86]. The study revealed extensive alterations in miRNA profiles associated with age and AD pathology. Notably, several miRNAs were significantly differentially expressed between male and female mice at various stages of AD-type pathology progression in this mouse model (Figure 1).

Several miRNAs (miR-137, miR-181c, miR-9, miR-29a, and miR-29b-1) have been found to participate in the ceramide biosynthesis pathway, regulating rate-limiting enzyme serine palmitoyltransferase (SPT) in the brain [88]. Increased levels of ceramide have been associated with AD and other neurodegenerative diseases [121,122]. It was revealed that expressions of miR-137, miR-181c, miR-29a, and miR-29b-1 (but not miR-9) were downregulated, while SPTLC1/2 (SPT long chain 1/2) protein expression levels increased in female mice [88]. Furthermore, miR-137, miR-181c, and miR-29b-1 were downregulated in the serum of female mice relative to male mice [87]. Nevertheless, whether these miRNAs show sexually dimorphic expression in the human brain is currently unknown.

There are multiple relevant human studies defining sexually dimorphic changes in miRNAs in people with ALS, PD, and frontotemporal dementia (FTD) (Figure 2). In ALS, a set of miRNAs (miR 206, miR-133a, miR133b, miR27a, miR-155, miR-146a, and miR-221) were significantly more upregulated in males compared to females [92]. The levels of several miRNAs also exhibited gender biases in PD [93]. Specifically, miRNA-34a-5p and miR-146a-5p were significantly upregulated in men compared to women. Interestingly, these expression patterns were observed in PD patients who were never treated with levodopa [93]. Another study conducted in PD patients indicated downregulation of the miR-29 family (miR-29a, miR-29b, miR-29c) in the serum of PD patients compared to controls, and these levels were significantly more downregulated in males compared to females [94]. In a separate study involving circulating miRNAs in humans, it was observed that plasma miR-663a, miR-502-3p, and miR-206 were significantly reduced in FTD patients compared to healthy controls [95]. However, when analyzed by sex, miR-206 was only found to be significantly downregulated in male patients, while its levels in females remained comparable to healthy controls. The same study revealed that there was a significant difference in Let-7e-5p levels when comparing female FTD patients with healthy controls, and no difference was found in males or in the overall population. In an attempt to discriminate FTD from other neurodegenerative diseases, an additional plasma miRNA biomarker, miR-127-3p, was proposed [123]. Plasma miR-127-3p was able to help distinguish FTD from healthy controls and AD patients; however, no difference between males and females was detected.

Neurodegenerative disease and cognitive impairment are hypothesized to have cross-talk with neuroinflammation. For example, sepsis can induce severe and long-term cognitive deficits affecting memory, attention, verbal fluency, and executive function [124,125], and results in worse outcomes in older AD and AD-related dementia (AD/ADRD) patients [126]. Sex-biased miRNA expression patterns were reported in the hippocampus of a sepsis mouse model [89]. While miR-320 was downregulated in female old mice (24 months) on day 1 and increased on day 4 following treatment, no change was reported in males. Interestingly, X-linked miRNAs responded differentially in male and female brains. Particularly, miR-222-3p, miR-221-3p, and miR-652-3p were downregulated in both young and aged females 1 day following treatment. By contrast, three different X-linked miRNAs, miR-223-3p, miR-98-3p, and miR-662-5p, were upregulated in older males.

## 4. Sex-Biased miRNAs in Cerebral Vascular Diseases

The prevalence, symptoms, treatment responses, and outcomes of strokes are known to have strong sex-bias [127,128]. MiRNA regulation has been demonstrated to make contribution to the sex differences in response to stroke. In a study of sex-biased vulnerability to cerebral ischemia [75], it was found that miR-23a targets an X-linked inhibitor of apoptosis (XIAP). XIAP, encoded on the X chromosome, is a member of the potent endogenous human inhibitor of apoptosis (IAP) family [129]. XIAP was shown to contribute to the sex difference following cerebral ischemia, where it was reduced significantly in females post-ischemia [75]. Concomitantly, miR-23a was significantly increased in the female brain in response to cerebral ischemia. Thus, it was suggested that enhanced miR-23a expression is likely to be responsible for the decreased XIAP in females [75]. Sex-biased miRNA expression was reported in another cerebral ischemia study [90]. The study found that many miRNAs were induced following ischemia in the cortices of both male and female mice. However, several miRNAs exhibited sex differences in the degree of changes as well as the direction of changes. For example, miR-509-3p levels were increased in males and decreased in females, and miR-883b-3p levels were decreased in males and elevated in females [90]. Florijn et al. have shown that several X-linked and estrogen-regulated miRNAs were expressed in the neurovascular unit and may play a role in sex-specific responses to ischemic stroke [130]. However, it remains speculative as to whether these miRNAs exhibit a sexually dimorphic response to ischemic stroke, and these issues will need to be further examined.

## 5. Sex-Biased miRNAs in Traumatic Brain Injury (TBI)

TBI is defined as any external impact that results in brain function alteration [131]. TBI is a major cause of death and disability [132]. According to the CDC, about 190 Americans died from TBI-related injury each day in 2021. For those who survive the immediate primary injury, there may be a secondary injury (which may take days or months) consisting of complex molecular and biochemical responses [133,134,135,136,137]. Moreover, there is a clear association between a history of TBI and an increased risk of developing neurodegenerative diseases, including AD, later in life [138,139,140,141,142,143,144], suggesting there may have overlapping pathophysiological pathways between TBI and neurodegenerative diseases [145,146,147].

Numerous studies suggest a sexual dimorphism in TBI responses (see details in review by Gupte et al., 2019). Men are three-times more likely than women to die from TBI, and women often have worse symptom burdens and take longer time to recovery [31,32,33,34,35,36,37]. Multiple factors impact the outcome measurement of a TBI. For example, when studying gender difference, the injury severities and the circumstances of brain injury—whether civilians injured in a motor vehicle accident, military personnel experienced blast-induced TBI, or athletes developed chronic traumatic encephalopathy following repeated head injury—are the primary factors in the outcome of TBI. Other biological factors (age, sex hormones, injury characteristics, etc.) and extrinsic factors (size of a study cohort, clinical or physiological markers used in the study, social and behavioral elements) can all contribute to the sex-biased outcomes following TBI. Although it remains debated [32,148,149,150,151], human studies often report worse experiences (recovery, cognitive function, overall quality of life) in women following TBI [31,32,33,34,150,152]. Nevertheless, recent reports in sport-related concussion (SRC) found no differences overall in return-to-play time between male and female collegiate athletes [149], although females in contact and males in limited-contact sports experienced longer times in recovery [149]. In contrast, animal studies often present a different narrative in that better outcomes were reported more in female animals [153,154,155,156]. These mixed observations further underscore the importance of identifying sex/gender interacting cofounders in a given study, and translating animal data to human clinical studies should be performed with caution.

MiRNAs are widely altered following TBI and are involved in various pathophysiological processes, such as neuroinflammation in the injured brains [157,158,159,160]. Clinical studies using extracellular biofluids have suggested that miRNAs are abundantly present in biofluids and may serve as biomarkers for diagnosis, prognosis, and distinguishing TBI subtypes (e.g., mild vs. severe) [161,162,163,164,165,166,167]. For example, extracellular vesicle miRNAs are differentially associated with TBI history in military personnel [161]. Interestingly, the altered miRNAs in biofluids were found to be commonly implicated in neuroinflammation, apoptosis, vascular remodeling, BBB integrity, and cellular repair pathways [161,162,163,164,165,166,167]. Despite the sex-biased nature of TBI response, past studies, especially experimental TBI and TBI-related clinical trials, have been largely focused on male animals and men [32,168,169,170,171], whereas only limited prior studies have reported on miRNA sex-related differences in TBI. Some studies have reported no difference in miRNA expression profiles between males and females [158]. Nevertheless, it is encouraging that gender/sex has now been integrated as a biological variable in many recent TBI-related miRNA clinical studies [31,32,33,34,35,36,37,38,148,149,150,151,152].

Studies by Kodama et al. have shown that miRNAs were expressed differently by sex, which influences function in mouse microglia, the major immune cells in the brain; furthermore, these differences were more pronounced in aging brains [91]. Sex-specific alterations were recently reported in several inflammatory miRNAs in brain CD11b+ cells isolated from naïve and TBI mice [98]. Among the tested inflammatory miRNAs (miR-146-5p, miR-150-5p, miR-155-5p, and miR-223-3p), higher levels of miR-150-5p and miR-155-5p were observed in male brain CD11b+ cells isolated from naïve animals. Interestingly, the response of these inflammatory miRNAs to TBI was distinct. For example, miR-155-5p was elevated in the brain CD11b+ cells of both female and male mice, but female mice showed a greater induction at an early time point (3 h) compared to male mice following TBI. On the other hand, miR-150-5p levels only increased in female mice and at more chronic time points of 7 and 14 days following TBI. Despite no significant sex-biased expression having been demonstrated with miR-146a-5p and miR-223-3p in CD11b+ cells isolated from naïve mice, these miRNAs also exhibited sex-biased responses following TBI. Late increased levels of miR-146a-5p were demonstrated in females relative to male CD11b+ cells following TBI. The levels of X-linked, anti-inflammatory miR-223-3p were markedly elevated in the brain CD11b+ cells of both sexes at 3 and 24 h following TBI, with the levels in females being significantly higher than those males at 24 hr. Further analysis has demonstrated a corresponding greater reduction in miR-223-3p-validated targets TRAF6 and FBXW7 in female relative to male CD11b+ cells. These data suggest that miR-223-3p and other inflammatory-responsive miRNAs may play key roles in sex-specific neuroinflammatory response following TBI.

In an ionizing-radiation-induced brain injury mouse model, a study showed that miRNA changes were sex-biased, with larger changes observed in females than males [72]. Specifically, the miR-29 family members miR-29a and miR-29c were shown to be exclusively downregulated in female mice’s frontal cortices at 6 and 96 h post-exposure in response to ionizing radiation. The study further confirmed that the protein levels of a miR-29 target, DNA methyltransferase 3a (DNMT3a), were correspondingly upregulated. This suggests that in response to environmental cues and insults, sex-specific changes in miRNAs and their target proteins may undergo sex-dependent changes, influencing the degree of injury and outcome in males and females.

## 6. Impacts of X Chromosome-Linked miRNAs

Genetic background and hormonal profiles are markedly different in males and females, which significantly impacts the expression and function of miRNAs and their machinery [50,172,173,174]. The human X chromosome contains roughly 10-times more protein genes compared to the Y chromosome. Accordingly, the human X chromosome contains higher numbers of miRNAs compared to the Y chromosome and autosomes [175,176,177]. There are currently 118 annotated miRNA genes on the human X chromosome and 62 mature miRNAs have been confirmed with high confidence (source: miRBase version 22.1) [176,178]. It is predicted that the human Y chromosome contains only four miRNA genes. No miRNA is identified on mouse or rat Y chromosomes. For the readers convenience, we have compiled a full list of the currently annotated human, mouse, and rat X-linked miRNAs in Appendix A (source: miRbase version 22.1).

The difference in X chromosome gene dosage between females (two X chromosomes) and males (one X chromosome) is balanced by a mechanism known as X-chromosome inactivation (XCI) [179,180,181]: one copy of the two X chromosomes in females is randomly and permanently inactivated during early embryogenesis. Nevertheless, about 15–25% of human X-linked genes escape inactivation to some degree, which is linked to some diseases [182,183,184,185,186]. A number of miRNAs reside in the intronic regions of their host coding genes that are known to escape XCI. This suggests that the intronic miRNAs residing in XCI-escaped genes can potentially be reactivated [97,187]. However, very few examples of X-linked miRNA escape have been experimentally verified. Data analysis has revealed that miR-548am-5p might potentially escape XCI, and it was experimentally verified to express significantly higher levels in human primary dermal fibroblasts of females relative to males [97]. Remarkably, it has been shown that the immunity-related genes in the inactive X chromosome in female lymphocytes are susceptible to being partially reactivated, resulting in a higher level of immunity-related gene expression [188]. This raises the possibility that immune-related miRNAs (such as miR-223-3p) could escape XCI, especially upon altered physiological (e.g., development, aging) or pathological (e.g., cell injury, inflammation, infection) conditions [85,101,176,189,190,191].

The expression of male-biased miRNAs appears to preferentially enrich for X-linked miRNAs [83,192]. Ro et al. have reported that ~39% of testis-specific or testis-preferential miRNAs are X-linked miRNAs [193], suggesting a potential role of X-linked miRNAs in spermatogenesis [193,194,195]. Several studies have shown that many X-linked miRNAs escape meiotic sex chromosome inactivation (MSCI) and suggest that these escaped miRNAs may be specifically required during the meiotic stages of spermatogenesis in males [195,196,197,198]. However, recent work by Royo et al. has argued that X-linked miRNA expressions are highly stage-dependent during MSCI and that the observed ‘MSCI escaped’ X-linked miRNAs may be a result of long half-life of the miRNAs [194].

Dysregulation of X-linked miRNAs is also associated with a number of diseases and pathological conditions [77,199]. The X-linked miRNA, miR-223-3p, has been demonstrated to play critical roles in immunity, inflammation, and lipid metabolism in various disease conditions [200,201,202,203,204,205]. Although no sex-biased expression is reported under physiological conditions, it has been shown that miR-223-3p exhibits sex-specific alterations in mouse brain CD11b+ cells in response to TBI [98]. Many other X-chromosome-located miRNAs (including miR-18b, miR-19b, miR-20b, miR-106a, miR-221, miR-222, miR-424, and miR-503) have not been investigated in the context of sexual dimorphism but potentially could provide additional understanding in the mechanism of sex differences in disease phenotypes. It is noteworthy that two X-linked miRNAs, miR-424 and miR-503, are members of the miR-15/107 group [206], which has been implicated in regulating key proteins (BACE1, GRN) in neurodegenerative diseases [119,207,208]. However, these miRNAs have yet to be investigated for their roles in sex-biased diseases.

## 7. Conclusions and Perspectives

MiRNAs are critical gene mediators in development and post-development processes as no known multi-cellular organisms can sustain viability and functionality when missing essential miRNAs or miRNA machinery [49,209,210,211]. Moreover, mutations in miRNA genes or the miRNA target site of a gene often cause diseases [212,213,214,215,216,217,218,219]. However, the functional role of sex-biased miRNAs in neurological diseases and neurotrauma is underappreciated and understudied. Many questions remain unanswered. For example, in an aged brain where sex hormones have declined, what drives sex-biased miRNA expression? Which X-linked miRNAs escape XCI, and under what circumstances? Do human and experimental animal models express the same sex-biased miRNAs in the brain? If not, how can we better implement knowledge of animal sex-biased miRNAs in human studies?

Other important issues include considering sex-biased miRNA trafficking between peripheral tissues and the CNS. Neurological diseases are a diverse group of disorders associated with a variety of pathological and dysregulated physiological processes. Peripheral events, physiological or pathological, such as inflammation, diabetes, hypertension, microbiome, and endocrine disorders, can have significant impacts on the CNS. In this review, we mainly focused on the brain’s sex-biased miRNAs relevant to neurological disease. However, sex-biased expression and response miRNAs in peripheral tissues may also play an important role in the physiology and pathology of the brain, including immune response and inflammation. Although the brain possesses ‘immune privilege’ to prevent damage mediated by inflammation [220], it is now clear that there is a dynamic immune and inflammatory interaction between peripheral tissues and the brain, especially when the brain undergoes pathological changes or is under abnormal conditions [221]. Considering that neuroinflammation is a common component of neurological diseases, peripheral sex-biased inflammatory miRNA trafficking may be an important element dictating brain responses to inflammation. Accumulated evidence has demonstrated that miRNAs can be transported between brain and peripheral tissues via extracellular vesicles as well as other protein complexes that readily cross the blood–brain barrier and exert functional roles [222,223,224]. This evidence has also led to the novel development of using miRNAs as a biomarker and therapeutic strategy for neurological diseases [225,226].

Given the broad impacts of miRNAs in normal neuronal physiology and pathology, sex differences in miRNA functions have important implications for assessing the risk of neurological disease, defining therapeutic intervention strategies, as well as in basic research and clinical investigations. The relative paucity of extant sex-related miRNA studies and reports in biomedical research is a limitation of the field and should be addressed.

## Figures and Tables

**Figure 1 ijms-25-02648-f001:**
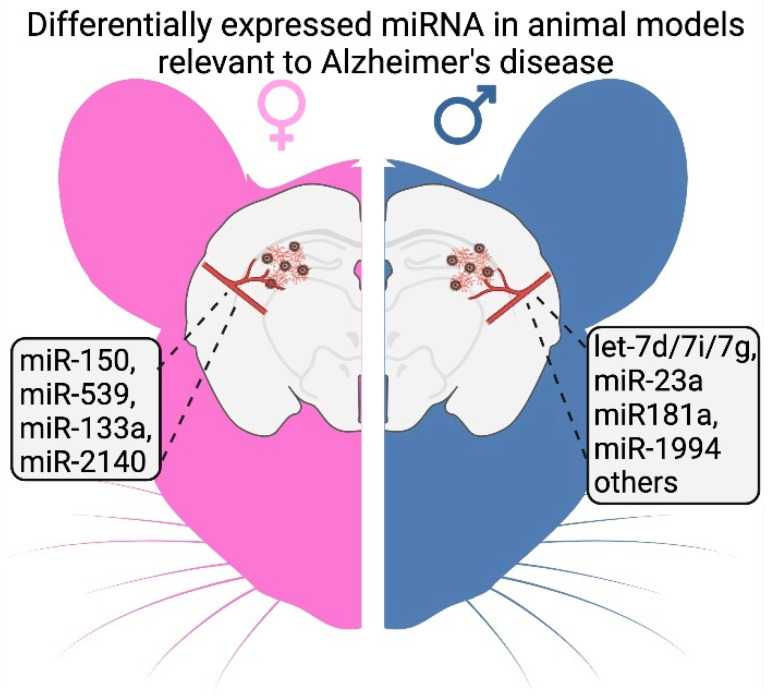
Example of sex-dependent miRNA expression in the 3xTg-AD mouse model. Chum et al. [86] analysis of 599 miRNAs in 3xTg-AD mice brain and cerebral vessels, showing sex-specific expression changes with age and degree of pathological progression.

**Figure 2 ijms-25-02648-f002:**
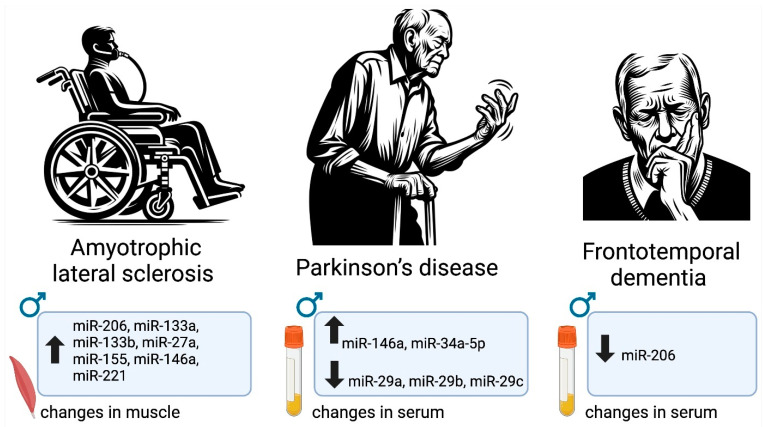
Sex-specific miRNA changes reported in men with ALS, PD, and FTD. In ALS, a specific set of miRNAs including miR-206 and miR-146a exhibited higher upregulation in males [92], while in PD, miRNAs like miR-34a-5p showed a similar male bias [93]. Conversely, in FTD, certain miRNAs such as miR-206 were significantly downregulated only in males [95].

**Table 1 ijms-25-02648-t001:** Sex-biased miRNAs in neurological diseases.

	miRNAs	Species/Tissue	Disease/Condition	X-Linked Yes/No	Observation	Reference
Pre-clinical studies	miR-200 family (miR-141-3p, miR200a-3P, miR200b-3P, and miR-429), miR-875	Rat/cortex	P0/development	No	Upregulated in females	[85]
	miR-200 family (miR-141-3p, miR200a-3P, miR200b-3P, and miR-429), miR-875	Rat/cortex	P7, adult/development	No	Upregulated in males	[85]
	miR-935	Rat/cortex	P0, P7, adult		Upregulated in males	[85]
	miR-322, miR-574, and miR-873	Mouse/brain	Prenatal stress	No	Downregulated in males	[80]
	Let-7g, miR-1944	Mouse/cerebral vessels	3xTg-AD, young to CI	No	Downregulated in males	[86]
	miR-133a, miR-2140	Mouse/cerebral vessels	3xTg-AD, young to CI	No	Downregulated in females	[86]
	miR-99a	Mouse/cerebral vessels	3xTg-AD, CI to Aβ	No	Downregulated in males	[86]
	let-7d, let-7i, miR-23a, miR-34b-3p, miR-99a, miR-126-3p, miR-132, miR-150, miR-151-5p, miR-181a	Mouse/cerebral vessels	3xTg-AD, pre-AD to AD	No	Changed in males	[86]
	miR-150, miR-539	Mouse/cerebral vessels	3xTg-AD, pre-AD to AD	No	Changed in females	[86]
	miR-137, miR-181c, miR-29a, miR29b-1	Mouse/brain	C57/BL	No	Downregulated in females	[88]
	miR-137, miR-181c, miR29b-1	Mouse/serum	High-fat diet/C57/BL	No	Downregulated in females	[87]
	miR-320	Mouse/hippocampus	Sepsis/old mice	No	Changed in females not males	[89]
	miR-223-3p, miR-98-3p, and miR-662-5p	Mouse/hippocampus	Sepsis/old mice	Yes	Upregulated in males	[89]
	miR-23a	Mouse/brain	Cerebral ischemia	No	Upregulated in females	[75]
	miR-509-3p	Mouse/cortices	Cerebral ischemia	No	Upregulated in males downregulated in females	[90]
	miR-883b-3p	Mouse/cortices	Cerebral ischemia	No	Upregulated in females downregulated in males	[90]
	miR-142a-5p and 25 others	Mouse/microglia	B6C3F1/J/naive	Yes to some	Enriched in females	[91]
	miR-1298-5p and 60 others	Mouse/microglia	B6C3F1/J/naive	Yes to some	Enriched in males	[91]
	miR-150-5p, miR-155-5p	Mouse/CD11b+	C57BL/6J/naïve	No	Upregulated in males	[98]
	miR-150-5p, miR-155-5p, miR-146a-5p, miR-223-3p	Mouse/CD11b+	C57BL/6J/TBI	No, except miR-223-3p	Sex-specific response at different time points following TBI	[98]
	miR-29a, miR-29c	Mouse/brain	Radiation-induced brain injury	No	Upregulated in female	[72]
Clinical studies	miR-206, miR-133a, miR-133b, miR-27a, miR-155, miR-146a, miR-221	Human/muscle	ALS	No, except miR-222	Upregulated in males	[92]
	miR-146a, miR-34a-5p	Human/serum	PD	No	Upregulated in men	[93]
	miR-29a, miR-29b, miR-29c	Human/serum	PD	No	Downregulated in men	[94]
	miR-206	Human/plasma	FTD	No	Downregulated in men	[95]
	miR-30b	Human/cortices	Schizophrenia	No	Downregulated in females	[96]
	miR-548am-5p	Human/primary dermal fibroblasts	XCI escape	Yes	Upregulated in females	[97]

## Data Availability

Not applicable.

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
