# Peer review of "Sex-Biased Expression and Response of microRNAs in Neurological Diseases and Neurotrauma"

_ijms, 2024, doi:10.3390/ijms25052648_

Round 1
Reviewer 1 Report
Comments and Suggestions for Authors
This review of sex differences in miRNA is comprehensive and has inspired both clinical and basic researchers to produce more evidence in the fields of neurological disorders and neurotrauma. However, there are a few inconsistent statements that require further confirmation, especially in the TBI section. In the introduction, the authors mentioned that TBI in females requires longer recovery times. However, lines 259-261 present a different statement. The symptom burdens and recovery might vary depending on the context of brain injury (athletes, military personnel, or civilians). A recent study suggested no difference in recovery (but not symptoms burdened) among athletes with concussion (NCAA, A CARE Consortium Study 2023). Adding some discussion around this context might be beneficial.
Additionally, it might be beneficial to organize Table 1 based on preclinical and clinical samples (rats, mice, humans) instead of rats, human, mice. Over the past decade, numerous studies on clinical TBI using miRNAs have been published, many of which were funded by the Department of Defense. It would be advisable to include some of these clinical studies in the TBI section.
Author Response
Comments and Suggestions for Authors
Comments: This review of sex differences in miRNA is comprehensive and has inspired both clinical and basic researchers to produce more evidence in the fields of neurological disorders and neurotrauma.
Response: We thank the reviewer for the positive comments.
However, there are a few inconsistent statements that require further confirmation, especially in the TBI section. In the introduction, the authors mentioned that TBI in females requires longer recovery times. However, lines 259-261 present a different statement. The symptom burdens and recovery might vary depending on the context of brain injury (athletes, military personnel, or civilians). A recent study suggested no difference in recovery (but not symptoms burdened) among athletes with concussion (NCAA, A CARE Consortium Study 2023). Adding some discussion around this context might be beneficial.
Response: We appreciate the reviewer for pointing out these inconsistences. We have revised both “Introduction” and the main text to further clarify the mixed observations of sex-biased phenomenon in TBI. We also added additional references to support the statements made in the manuscript.
In Introduction section:
Original: “According to the Centers for Disease Control and Prevention (CDC), men are two times more likely to be hospitalized and three times more probable to death due to TBI, yet women were found to require longer to recover from a TBI [31-34].”
Revised: “According to Centers for Disease Control and Prevention (CDC), men are two times more likely to be hospitalized and three times more probable to death due to TBI, yet women were more vulnerable than men to persist TBI-related cognitive and somatic symptoms [31-34]. Moreover, a longer time to symptom resolution was reported in high school and collegiate female athletes [35-38].”
In TBI section:
Original: “Although it remains debated [144], numbers of studies suggest that women seem to have a better recovery from all TBI (mild to severe), and both human and animal studies of more severe TBI report better outcomes in females than males [32, 145, 146]. Therefore, it is critical to incorporate sex/gender in clinical and basic TBI research.”
Revised: “…., and women often have worse symptom burdens and take longer time to recovery [31-37]. Multiple factors impact an outcome measurement of a TBI. For example, when study gender difference, the injury severities, and the circumstances of brain injury—whether civilians injured in a motor vehicle accident, military personnel experienced blast-induced TBI, or athletes developed chronic traumatic encephalopathy following repeated head injury--, are the primary factors in the outcome of TBI. Other biological factors (age, sex hormones, injury characteristics etc.) and extrinsic factors (size of a study cohort, clinical or physiological markers used in the study, social, and behavioral elements) can all contribute to the sex-biased outcomes following TBI. Although it remains debated [32, 149-152], human studies often report worse experiences (recovery, cognitive function, overall quality of life) in women following TBI [31-34, 151, 153]. Nevertheless, recent reports in sport-related concussion (SRC) found no differences overall in return-to-play time between male and female collegiate athletes [150], although females in contact and males in limited contact sports experiencing longer time in recovery [150]. In contrast, animal studies often present a different narrative, in that, better outcomes were reported more in female animals [154-157]. These mixed observations further underscore the importance of identifying sex/gender interacting cofounders in a given study and it should be with caution when replicate animal data to human clinical studies.”
Comments: Additionally, it might be beneficial to organize Table 1 based on preclinical and clinical samples (rats, mice, humans) instead of rats, human, mice.
Response: we appreciate the reviewer for this valuable suggestion. In response, we have re-arranged Table 1 based on preclinical and clinical studies.
Comments: Over the past decade, numerous studies on clinical TBI using miRNAs have been published, many of which were funded by the Department of Defense. It would be advisable to include some of these clinical studies in the TBI section.
Response: we appreciate the suggestion and have included following phrases and references. Nonetheless, we did not find sex-specific miRNA investigations in these studies.
Revised (added phrases): “Clinical studies using extracellular biofluids suggested that miRNAs are abundantly present in biofluids and may serve as biomarkers for diagnosis, prognosis, and distinguishing TBI subtypes (e.g. mild vs severe) [162-168]. For example, extracellular vesicle miRNAs differentially associated with TBI history in military personnel [162]. Interestingly, the altered miRNAs in biofluids were found to commonly implicated in neuroinflammation, apoptosis, vascular remodeling, BBB integrity, and cellular repair pathways [162-168]. … Nevertheless, it is encouraging that gender/sex is now integrated as a biological variable in many recent TBI-related miRNA clinical studies [31-38, 149-153].”

Reviewer 2 Report
Comments and Suggestions for Authors
The authors present a good review that exposes the alterations in microRNAs potentially associated with the observed differences between males and females in the prevalence of neurodegenerative, neurological, and neurotraumatic conditions. They outline the variations in these non-coding RNAs across various pathologies. Additionally, they incorporate a section analysing how certain microRNA disparities may be related with genes located on the X chromosome that evade inactivation. The manuscript offers a comprehensive overview of pertinent information on the topic.
Finally, the review underscores the significance of incorporating both genders in studies related to neurological diseases, basic or applied research, in the search for biomarkers or therapeutic targets.
Author Response
Reviewer 2
Comments: The authors present a good review that exposes the alterations in microRNAs potentially associated with the observed differences between males and females in the prevalence of neurodegenerative, neurological, and neurotraumatic conditions. They outline the variations in these non-coding RNAs across various pathologies. Additionally, they incorporate a section analysing how certain microRNA disparities may be related with genes located on the X chromosome that evade inactivation. The manuscript offers a comprehensive overview of pertinent information on the topic.
Finally, the review underscores the significance of incorporating both genders in studies related to neurological diseases, basic or applied research, in the search for biomarkers or therapeutic targets.
Response: We appreciate the reviewer for these positive remarks.

Round 2
Reviewer 1 Report
Comments and Suggestions for Authors
Thanks for the revision. I have no further comments.